# E-Cadherin Expression in Relation to Clinicopathological Parameters and Survival of Patients with Epithelial Ovarian Cancer

**DOI:** 10.3390/ijms232214383

**Published:** 2022-11-19

**Authors:** Michal Kielbik, Izabela Szulc-Kielbik, Magdalena Klink

**Affiliations:** Institute of Medical Biology, Polish Academy of Sciences, Lodowa 106, 93-232 Lodz, Poland

**Keywords:** E-cadherin, ovarian cancer, patients survival, clinicopathological parameters

## Abstract

It is generally accepted that loss/reduction of E-cadherin expression on tumor cells promotes their migration, invasiveness, and metastasis. It is also an indicator of cancer cells’ aggressiveness. The aim of this study was to assess how the expression of E-cadherin varies in primary ovarian cancer tissue in regard to overall survival of patients; FIGO stage; grade; histopathological type of tumor; and potential factors discriminating malignant and nonmalignant ovarian tumors. Our analysis was based on literature research (1 January 2000–8 November 2021) conducted according to the PRISMA guidelines. Most studies support the assumption that loss/reduced expression of E-cadherin results in shorter overall survival of EOC patients. Moreover, most research has shown that there is a correlation between the low level of E-cadherin and the advancement stage of disease, especially in high-grade serous ovarian carcinoma type. However, E-cadherin expression seems to not be helpful to distinguish malignant and nonmalignant tumors. In conclusion, reduced E-cadherin expression in primary ovarian cancer tissue may indicate a less favorable disease outcome and is associated with high advancement of the disease.

## 1. Introduction

### E-Cadherin in Cancers

E-cadherin is a calcium-dependent adhesion molecule belonging to the superfamily of cadherins. It is a type I classical cadherin encoded by the *CHD1* gene located on chromosome 16q22.1. Since its identification and characterization was noted as early as 1977 by Takeichi, it is generally considered the prototype of all cadherins [1]. E-cadherin is a 120-kDa transmembrane glycoprotein that is subdivided into three domains: extracellular (ectodomain), transmembrane, and intracellular (cytoplasmic). The extracellular domains join with the cytoplasmic tail and create signaling hubs called adherens junctions (AJs), which mediate most cadherin interactions. The extracellular domain is composed of five tandemly repeated cadherin subdomains, also called extracellular cadherin repeats (ECs). The five ECs mediate Ca-dependent interactions with cadherin subdomains on adjacent cells and provide homotypic cell-to-cell interplay. The cytoplasmic tail is another very important component of this adhesion molecule. It is subdivided into the membrane proximal cytoplasmic domain, also known as the juxtamembrane domain, and the (β)-catenin binding domain (CBD), which binds to (β)-catenin. This cadherin–catenin complex associates with α-catenins, while homodimeric forms a α-catenins link with F-actin, binding the entire cadherin molecule with the cytoskeleton. The whole E-cadherin–catenin complex is crucial in maintaining epithelial cell integrity and cell–cell adhesion. Moreover, apart from maintaining tissue architecture, E-cadherin expression affects crucial stages of embryogenesis and organogenesis [2,3,4,5].

E-cadherin is most widely known as a tumor suppressor because it prevents the dissociation of cells from the tumor mass, thus inhibiting their migration and metastasis. Any dysfunction of the E-cadherin–catenin complex, such as downregulation or loss of E-cadherin expression, results in an acquisition of the mesenchymal phenotype of tumor cells. This is most often associated with an increase in metastatic potential [4,6,7]. There are several mechanisms that are implicated in E-cadherin inactivation/loss in different types of human carcinomas. One is the loss of heterozygosity of chromosome 16q21-22, which is frequently accompanied by various somatic mutations in the *CHD1* gene (e.g., skipping of exon 7 or 9). E-cadherin becomes inactive as a result of these mutations, which has been described for breast, gastric, endometrial, ovarian, and thyroid cancers [3,8,9]. The expression of E-cadherin is also controlled on the epigenetic level. One of the important epigenetic events that occurs during cancer progression is hypermethylation of the CpG-rich region in the 5′ proximal promoter of the E-cadherin gene, leading to the downregulation of the E-cadherin gene [3,10]. Another mechanism causing loss of E-cadherin expression relies on the activity of numerous transcription factors, such as SNAIL-1, SLUG, TWIST, deltaEF1/ZEB1, and SIP/ZEB2. These transcriptional repressors negatively modulate E-cadherin transcripts by binding to the E-box elements of its promoter. The exact mechanism may differ for each transcription factor; for instance, SNAIL/SLUG recruits complexes consisting of histone deacetylates (HDAC1, HDAC2) and transcription regulatory protein (SIN3A), while TWIST engages histone–lysine N-methyltransferase, but in all cases, it leads to the transcriptional repression of the E-cadherin promoter [11,12]. The outcome of this process is the so-called “cadherin switch”, defined as a significant drop in E-cadherin expression with a simultaneous increase in N-cadherin expression [13]. This kind of association between the level and activity of those transcriptional repressors and the expression of E-cadherin has been confirmed and documented in colorectal, breast, and gastric cancer models [3,10,14,15,16]. There is also an epigenetic mechanism associated with the expression of specific microRNAs. A high level of miRNA-10b downregulates E-cadherin expression at the posttranscriptional level in nasopharyngeal carcinoma [17].

In addition to processes that regulate E-cadherin gene expression, there is also a great variety of mechanisms focused on posttranslational modification of the E-cadherin protein. One of them includes endocytosis and proteolytic processing as an alternative way of inhibiting the regular function of E-cadherin. Under normal conditions, this adhesion molecule undergoes clathrin-dependent endocytosis and is then recycled to a new site in the plasma membrane to form new cell–cell contacts [3,5,18]. However, the abnormal activation of proto-oncogenes such as *Met* or *Src*, which is observed in cancer cells, leads to the phosphorylation of tyrosine residues in the conserved domain in the cytoplasmic tail of E-cadherin, resulting in ubiquitin-dependent degradation of this molecule [3]. Moreover, various metalloproteinases (MMP3, MMP7, MMP9, MMP14, ADAM10) cause shedding of the E-cadherin extracellular domain near the plasma membrane by proteolytic degradation of adherent junctions, which leads to downregulation of E-cadherin expression on the cell surface and inhibits normal cell–cell adhesion, thus inducing cell motility [3,10].

It is generally accepted that loss of E-cadherin expression on tumor cells promotes its migration, invasive potential, and metastasis, and is a hallmark of epithelial-to-mesenchymal transition (EMT), as has been demonstrated in various cancer cell lines and in animal models [9,19,20]. However, some data have questioned whether the loss of E-cadherin expression is a marker of cancer aggressiveness. It was found that tumor cells with a mesenchymal phenotype and undergoing metastasis can still express high levels of E-cadherin, and this adhesion molecule is not necessary for EMT [20,21,22]. Moreover, it was also described that both E-cadherin loss and EMT are not required for the motility, invasion, and metastasis of pancreatic cancer cell lines [23]. An interesting study showing that loss of E-cadherin even inhibited ovarian cancer movement and migration was presented by Choi et al. [24] in a model of 3D Matrigel culture of the OVCA432 ovarian cancer cell line. The spheroids formed by OVCA432 cells were characterized by high levels of E-cadherin and effective migration when transferred into the collagen I matrix. The knockdown of E-cadherin (siRNA) in OVCA432 cells resulted in the formation of a lower number of spheroids and inhibition of their movement in the collagen I matrix. On the other hand, many clinical studies on ovarian cancer [25,26], bladder cancer [27], breast cancer [28], and hepatocellular carcinomas [29] have shown that E-cadherin loss indicates an invasive characteristic of carcinoma, tumor progression, and poor prognosis of patient survival. Thus, despite the controversial results concerning the correlation of cell invasiveness with E-cadherin loss, many studies underline that the evaluation of E-cadherin levels in tumor masses may be a promising indicator of prognosis in cancer patients.

## 2. E-Cadherin Expression in Relation to Clinicopathological Parameters and Survival of EOC Patients

Since E-cadherin expression loss in cancer cells is strongly connected with tumor progression and a more aggressive phenotype of cells, as well as their worse response to chemical drugs, it opens up an opportunity to consider this protein as a marker of various clinicopathological parameters of ovarian cancer. Our review is based on the papers published during last the 20 years and its aim is to systematize the knowledge about the E-cadherin expression in the primary ovarian tumor tissue in relation to the clinicopathological parameters and epithelial ovarian cancer (EOC) patients’ survival. Moreover, our review brings a new inquisitive look in the worldwide discussion about the opportunity to consider E-cadherin expression loss as an independent biomarker in ovarian cancer.

We decided to choose and evaluate one simple parameter—E-cadherin protein expression in primary tumor samples tested by the immunohistochemistry method, which is commonly utilized in clinics on routinely sourced tissue samples. Analysis of this parameter should give a clear “yes” or “no” answer, without consideration of mechanisms/factors affecting E-cadherin appearing or loss in cells.

### 2.1. Literature Search and Selection Criteria

The literature search was based on the PRISMA guidelines and was conducted in 9 databases available in the Web of Science all Data Bases (MEDLINE, BIOSIS Citation Index, Current Content Connect, Derwent Innovations Index, Data Citation Index, KCI-Korean Journal Database, Russian Science Citation Index, SciELO Citation Index). We used the search terms “ovarian cancer and E-cadherin” in the time interval from the 1st of January 2000 until the 8th of November 2021.

Initially, we found 1696 articles, from which we excluded meeting abstracts, review articles, editorial materials, letters to the editor, books, and patents. Thus, 1261 papers were left from the following records: articles, other, data studies, case reports, and clinical trials. In the next step, we excluded articles containing key words: animals, mice, disease model, animals, and rats. Finally, 989 articles remained. Ultimately, after exclusion of articles in languages other than English, we reviewed the titles and abstracts of 930 potentially relevant studies. During their analysis, the exclusion criteria covered studies on the mRNA level of E-cadherin, methylation of E-cadherin, cell lines, gene databases, systematic reviews, meta-analyses, other tumor types, endometriosis, E-cadherin not related to ovarian cancer patients, animal models, and ovarian cancer cells isolated from tumors and tested in vitro. We included studies concerning (1) the expression of E-cadherin in primary ovarian cancer tissue in relation to clinical parameters (FIGO, grade, histopathological type of tumor) and patient survival and (2) the expression of E-cadherin in primary ovarian cancer tissue and benign ovarian tumors. We chose 82 articles for full text reading. However, 7 of them were initially rejected due to the availability of abstracts only. Furthermore, after full-text reading and analysis, we chose 46 papers to present in this review. The excluded studies from the read of 75 records covered E-cadherin expression only in correlation/combination with other biomarkers and/or tissue antigens; ovarian cancer type other than epithelial; expression of E-cadherin tested with Western blot method; expression of E-cadherin on cancer cells isolated from ascites and/or tissue; cell lines; and expression of E-cadherin in patients after treatment. Moreover, we excluded three articles in which E-cadherin expression was compared between primary ovarian tumors and metastatic lesions. The detailed search strategy, with inclusion and exclusion criteria, is presented in Figure 1.

### 2.2. Results

In all articles analyzed here, the expression of E-cadherin was determined with immunohistochemistry. Almost half of all studies (46%) examined only membranous localization of E-cadherin, while in 41% of the studies, E-cadherin’s immunoreactivity was not localized. Additionally, in one paper, separate data for membranous, cytoplasmic, and total (membranous + cytoplasmic) expression of protein were presented, whereas in another study authors showed only cytoplasmic E-cadherin immunoreactivity. Lastly, three studies evaluated the combined immunoreactivity of membranous and cytoplasmic E-cadherin, presented as total level of discussed adhesion molecules. Although the determination of E-cadherin expression was not unequivocal, all papers were published after the peer-review process, thus we decided to present here all studies independently of E-cadherin localization. However, the localization of E-cadherin immunoreactivity in cell is marked in all tables. Moreover, we would like to underline that after deep analysis of published results, we can conclude that membranous or not localized E-cadherin immunoreactivity was not connected with *p* value. In most papers, the expression was assessed using a semiquantitative scoring system based on the quantification of the percentage of positively stained cells (0–100%) and the intensity of immunoreactivity (0—none; 1—low; 2—moderate; 3—high). The cutoff was mainly accepted as ≥10% or ≥25% of stained cells and at least moderate intensity of staining (score 2 and 3). However, in some papers, 1% of stained cells was also interpreted as positive expression, while in some reports, only more than 50% of immunoreactive cells was recognized as positive staining. Moreover, in some articles, the quantification of E-cadherin expression was established with the percentage of stained cells only. Nevertheless, because all analyzed papers were successfully published (after review), we decided not to discriminate these with the low cutoff and/or lacking the intensity of the immunoreactivity parameter. In general, the data were published as a number (percentage) of patients, determined by authors as positive or preserved and negative or reduced/low expression of E-cadherin (calculated according to the semiquantitative scoring system) or as an immunoreactive score (IRS) being the sum of % of stained cells and intensity of immunoreactivity. Incidentally, the results were also presented as negative, reduced, weakly positive, and strongly positive or with the “plus and minus” score.

We evaluated the relationship of E-cadherin expression with (i) patients’ overall survival (OS) and progression-free survival (PFS); (ii) patients’ FIGO stage and grade, as well as the histopathological type (HP) of the tumor; and (iii) as a potential parameter helping to discriminate malignant and nonmalignant ovarian tumors.

#### 2.2.1. Association of E-Cadherin Expression with Patient Survival

A total of twenty papers considered the relationship of patient survival with preserved and/or reduced expression of E-cadherin. Three of them were finally rejected after deep analysis due to insufficient data [30,31] and compilation of ovarian cancer with borderline tumors [32].

Two papers [33,34] referred to EOC patients in advanced (FIGO III and IV) stages only. Both studies clearly indicated that preserved membranous expression of E-cadherin was associated with a longer OS. The calculated hazard ratio (HR) was 3.084, while the 95% confidence intervals (95% CI) were 1.541–6.175; *p* = 0.001; *n* = 54 in the study by Bačić et al. [34] and HR = 2.7; 95% CI = 1.3–5,9; *p* = 0.001; *n* = 98 in the study by Mise et al. [33]. Moreover, not localized [35,36] or membranous [37] E-cadherin expression was determined in the tumor tissue of patients with high-grade ovarian serous carcinoma (HGOSC). Two studies [35,37] showed that high E-cadherin immunoreactivity predicted improved survival, according to the Kaplan–Meier curve. The median OS of patients whose tumors displayed significantly higher E-cadherin expression in comparison to those whose tumors were characterized by reduced expression was 99 months versus 41 months, *p* = 0.043 [35] and 27 months versus 19 months, *p* = 0.008, respectively [37]. The number of enrolled HGOSC patients was relatively high and reached 98 and 177, respectively. In contrast, Song et al. [36] demonstrated no association of OS with the positive or negative expression of E-cadherin in HGOSC tissue (*n* = 198; follow up to 15 years). The HR was 1.245, and the 95% CI was 0.882–1.756 (*p* = 0.213). In the next two papers, enrolled patients represented serous ovarian cancer in all stages. Taskin et al. [38] showed that favorable survival was related to preserved total (membranous + cytoplasmic) E-cadherin expression. HR = 9.6; 95% CI = 2.1–43.6; *p* = 0.001; *n* = 30. However, Dian et al. [39] demonstrated no differences in the OS of patients according to not localized E-cadherin immunoreactivity, even in strong staining intensity (*p* ≥ 0.07; *n* = 100). In both studies, the semiquantitative scoring system included the percentage of stained cells and the intensity of immunoreactivity. Thus, it is difficult to conclude why these two reports evidenced opposite results.

In summary, five out of seven papers (71%) clearly indicate that preserved expression of E-cadherin is associated with favorable OS of ovarian cancer patients. However, it should be emphasized that the presented studies have some limitations. First, the HR and 95% CI were demonstrated only in four papers. Second, the enrolled ovarian cancer patients were strictly selected to the HGOSC, the serous histopathological type, or to the advancement of disease; thus, it is difficult to make a comparison between them.

The relationship of E-cadherin expression with the OS of EOC patients representing all stages, grades, and histopathological types was evaluated in nine articles. A summary of the obtained results is included in Table 1. The follow-up differed and amounted to 60 to 200 months. The number of enrolled cases per study ranged from 44 to 136. Five studies (according to the Kaplan–Meier test) clearly indicated that patients with primary ovarian cancer characterized by reduced expression of E-cadherin had significantly shorter OS than patients whose tumor tissue highly expressed E-cadherin [40,41,42,43,44]. In contrast, the Kaplan–Meier survival curve results in four studies [45,46,47,48] demonstrated that negative E-cadherin immunoreactivity did not determine poor OS in EOC patients. However, it should be noted that in the study described by Huang et al. [45], a score of >5% of stained cells was defined as positive expression, and the intensity of staining was not included, which could have influenced the result. As presented in Table 1, the HR and 95% CI calculations were available in only five papers. In the report by Voutilainen et al. [49], recurrence-free survival (RFS) was calculated. The data showed that preserved E-cadherin expression predicted favorable RFS, *p* = 0.038; *n* = 282, follow-up = 10 years.

In the case of the PFS parameter, two papers revealed that neither positive nor negative expression of E-cadherin was associated with PFS in EOC patients (*p* = 0.775; [48]) (*p* = 0.967; [47]). In contrast, Kim et al. [43] showed that overexpression of E-cadherin resulted in higher PFS in EOC patients (*p* = 0.03; *p* = 0.01, respectively) (Table 1).

In summary, most studies (6 out of 10; 60%) support the assumption that reduced/lost expression of E-cadherin results in shorter survival of EOC patients. The lack of statistical significance in the association of E-cadherin expression with patient OS in 40% of studies cannot be related to variance in the number of enrolled women or follow-up period, or cutoff value. However, it should be emphasized that in the two studies [47,48] showing no statistical association of reduced E-cadherin level with worse survival of patients, the intensity of protein’s immunoreactivity was not evaluated.

#### 2.2.2. Association of E-Cadherin Expression with Clinicopathological Parameters

##### FIGO Stage

The E-cadherin immunoreactivity profile regarding the advancement of epithelial ovarian carcinoma (all HP) was described in 15 articles. Two papers were rejected after the first analysis due to (i) enrollment of only FIGO III or IV stage patients [34] and (ii) matching FIGO II with FIGO III/IV patients [50]. As presented in Table 2, 9 of the remaining 13 articles showed that the tumor tissue of EOC patients with FIGO III and IV stages expressed significantly less E-cadherin than the tumor tissue of patients with early stages (FIGO I and II) of disease [25,26,43,51,52,53,54,55,56]. Nonetheless, four studies did not confirm the association of reduced E-cadherin expression with the advancement of disease and showed similar protein immunoreactivity in tumor tissue from patients in all cancer stages [32,49,57,58]. It should be noted that the results of one “negative” study are based on an extremely low number of patients included in the statistical analysis in each tested group: 5, 2, 12, and 0 for FIGO I, II, III, and IV, respectively [57]. Thus, in our opinion, this report should not be considered when drawing the conclusion.

Four subsequent articles that were included in our evaluation concerned patients with serous ovarian cancer only including both, HGOSC and low-grade ovarian serous carcinoma (LGOSC) [39,59,60,61]. Interestingly, four out of four (100%) articles showed that reduced tissue expression of E-cadherin was significantly associated with the advancement of serous ovarian cancer disease (FIGO III/IV stages) (Table 3).

In summary, most results (13 out of 17; 76%) showed that primary ovarian tumors of patients with advanced-stage ovarian cancer showed decreased expression of the characteristic epithelial marker E-cadherin. The lack of the association of reduced tissue immunoreactivity for E-cadherin with disease advancement in the remaining studies seems to be unrelated to cutoff value or membranous or cytoplasmic localization of immunoreactivity.

##### Grading

E-cadherin expression in relation to the tumor grade of EOC patients was the subject of 16 studies. Two of them were rejected after deep analysis due to the lack of clear information about statistical significance in the E-cadherin staining between grades 1, 2, and 3 of ovarian cancer [47] and the examination of its expression in relation to tumor grading and patient survival together [31]. A summary of the results described in the remaining 14 studies is presented in Table 4. Five studies showed that G3 ovarian cancer was characterized by significantly lower E-cadherin expression than the G1 and G2 types [25,32,49,52,54]. In contrast, six articles showed that positive as well as negative/reduced immunoreactivity of E-cadherin was at a similar level independent of ovarian tumor grading [43,51,53,55,57,58]. In the next 3 papers, the grade was estimated as low and high only, and the results also showed no difference in either E-cadherin positive or negative staining regarding tumor tissue differentiation [26,34,50]. In addition, seven studies estimated E-cadherin expression in relation to tumor grading in patients with serous ovarian cancer only (Table 5). The significant association of reduced E-cadherin immunoreactivity with high-grade serous carcinoma was revealed in four papers [39,61,62,63]. On the other hand, in three studies, no difference in E-cadherin expression between high-grade and low-grade serous OC was revealed [59,60,64].

In summary, about half of the published articles indicate an association of reduced E-cadherin expression with poorly differentiated ovarian tumor tissue, and the other half of papers show no relationship between tumor grade and E-cadherin expression. Thus, it is not possible to draw unequivocal conclusions. The opposite results may arise from, for example, high disproportionality in the number of EOC patients with G1, G2, and G3 tumors. In particular, the quantity in the G1 or low-grade tumor group was smaller than that in the G3 or high-grade group, which could influence the statistical examination (calculation). For example, in the study by Bačić et al. [34], the number of patients with low- and high-grade tumors amounted to 8 and 46, respectively.

##### Histopathological Type (HP)

Sixteen separate studies tested E-cadherin expression in primary ovarian tumors of different HP types. However, after detailed analysis of the results, five papers were rejected due to (i) a lack of clear information regarding the unit/score (percentage, intensity of staining) in which protein expression was calculated and/or presented [46,65,66]; (ii) enrollment of patients with serous and endometroid grade 3 types only [67]; and (iii) statistical analysis of E-cadherin immunoreactivity in relation to HP and patient survival together [31]. As presented in Table 6, the association of E-cadherin expression with the HP type of ovarian cancer is not unequivocal. In 6 out of 11 studies, no significant differences in the reduced or preserved expression of this adhesion protein among serous, mucinous, endometroid, or clear cell types were found [43,54,55,57,58,68]. However, in 4 out of 11 papers, the data showed that reduced expression of E-cadherin was linked with serous ovarian cancer [26,49,69,70]. One study showed that the endometroid type of ovarian cancer also had reduced expression of E-cadherin [49]. Moreover, two reports demonstrated that mucinous tumors were characterized by preserved expression of E-cadherin in a significantly higher number of patients [56,69].

According to the results presented above, we can conclude only that no clear relationship between E-cadherin expression and HP types of EOC is observed. Only in the case of serous carcinoma can its reduced expression be expected since it is evidenced in more than 30% of studies. The great limitation of these studies is the large difference in the cutoff value in the percentage of stained cells approved as positive, for example, ≥11% [58]; ≥1% [55]; and ≥25% [57]. Moreover, in some studies [55,56,58,69], the data were presented as negative, positive, and strong positive expression or preserved, reduced, and absent expression. However, in most studies, the expression was estimated as positive or negative only. We would like to point out that after deep analysis of the published results, we matched the data of reduced and absent expression as negative and weak and strong expression as positive, as this additional separation did not influence the conclusion. However, due to the importance of data, the results of the study by Yoshida et al. [56] were presented exactly as in their article.

#### 2.2.3. Can E-Cadherin Expression Be a Helpful for Discriminating Malignant and Nonmalignant Ovarian Tumors?

The E-cadherin expression in the association with malignant and nonmalignant ovarian tumors was the subject of 11 separate studies. Their detailed analysis resulted in the rejection of one paper due to the lack of information regarding E-cadherin expression estimation [65]. The results of the remaining 10 articles are presented in Table 7. Only four reports showed significantly higher staining of E-cadherin in benign tumors than in ovarian carcinoma [30,32,71,72]. In contrast, four papers [53,56,58,73] conclusively proved that E-cadherin expression did not differ between malignant and nonmalignant tumors. It should be noted here that Faleiro-Rodrigues et al. [73] estimated a very high cutoff value of stained cells (amounted ≥ 51%) regarded as positive/preserved expression. Moreover, interesting, two studies [45,68] revealed that a significantly higher percentage of EOC patients than benign ovarian tumor (BOT) patients showed positive expression of E-cadherin.

In summary, 40% of the studies indicated that E-cadherin expression does not differ between EOC and BOT patients, while 40% of the studies showed that EOC was characterized by reduced expression of E-cadherin in comparison to the BOT group. The greatest limitation of the studies assessing the viability of E-cadherin expression as a potential diagnosis biomarker is the lack of a strictly objective and standardized method for its estimation. Thus, according to the presented data, we can conclude that the tissue expression of E-cadherin cannot be a promising for discriminating malignant and nonmalignant ovarian tumors.

## 3. Discussion

This review summarizes the available data (20 years period) concerning E-cadherin expression in regard to the clinicopathological parameters and survival of EOC patients. First, we analyzed the relationship of E-cadherin expression with patient OS and PFS. Only three reports involved PFS evaluation, which revealed opposite results; thus, it is not possible to reach any real conclusion. Most published papers support the assumption that reduced expression of E-cadherin is associated with shorter OS in EOC patients, as well as in ovarian cancer patients selected for serous HP type or FIGO III/IV stage only. Thus, we suggest that reduced expression of E-cadherin can represent a potential risk factor for EOC patient survival. The decreased expression of E-cadherin has been evidenced as an important event of ovarian cancer progression because in 81% of analyzed articles, a significant association of low/reduced staining for this adhesion molecule with FIGO III/IV stages was evidenced. Interestingly, serous ovarian carcinoma of patients with FIGO III/IV stages was found to be negative for E-cadherin in all reports presented here. The contribution of E-cadherin to OS and tumor progression can be rationalized by its great involvement in the EMT process, which is preceded by the “cadherin switch” and the loss of E-cadherin. As a result, cells undergoing EMT change their phenotype from epithelial to mesenchymal and gain motility. This is widely known to be the first step in the cascade of tumor metastasis and can indicate a higher invasive potential of tumor tissue [9,22]. It is accepted that metastatic ovarian cancer has a five-year survival rate of approximately 39% (FIGO III) and 17% (FIGO IV) (https://ocrahope.org/patients/about-ovarian-cancer/staging/ accessed on 8 February 2022). It was also reported that low E-cadherin expression observed in the tumor tissue of patients in an advanced stage of disease would favor cancer cell dissemination [26]. However, it should also be emphasized that there are still many studies (see results presented above) that do not indicate any relation between E-cadherin positive/negative staining of tumor tissue and patient survival and/or disease advancement when metastatic lesions are present. It is worth remembering that loss of E-cadherin has been found to not be required and/or insufficient for EMT to occur. The question of whether this adhesion molecule is crucial and a major driver of EMT remains to be answered [22]. Thus, the correlation of E-cadherin positive and negative expression with cancer progression/metastasis as well as patients’ poor survival is not entirely obvious. On the other hand, it should be underlined that some authors [45,46,47,48] observed that a lack of correlation between E-cadherin reduced expression and worse survival of EOC patients can be connected with hardly unified scoring system to estimate protein immunoreactivity.

The analysis of relation of other clinicopathological features, HP, and grade with E-cadherin did not bring definite/unequivocal conclusions. The most promising are data regarding the association of low E-cadherin expression with G3 tumors that was observed in 50% of the studies presented here. Even in HGSOC patients (in all FIGO stages), only half of the reviewed studies demonstrated a correlation of this type of tumor with reduced E-cadherin expression. On the one hand, it is frequently found in the literature that poorly differentiated tumor cells (e.g., triple-negative breast cancer; colorectal cancer; endometrial cancer) are characterized by reduced expression of E-cadherin [74]. On the other hand, it was also found that ovarian cancer cells, especially the high-grade serous type, can express a hybrid phenotype with epithelial and mesenchymal markers simultaneously [75,76,77]. The analysis presented here showed that only 30% of studies demonstrated an association between reduced E-cadherin expression and serous HP ovarian cancer. It should also be emphasized that due to the multiple mechanisms that are involved in the downregulation of E-cadherin expression (reviewed in the Introduction section), the appearance or absence of this adhesion molecule in particular tumor tissue is strongly dependent on the multifunctional activity of the whole tumor microenvironment. The high presence of hepatocyte growth factor, epithelial growth factor, and transforming growth factor strongly affects the level and activity of transcriptional repressors (e.g., SNAILs), resulting in the induction of EMT [7]. Moreover, hypoxia, a hallmark of tumor tissue, attenuates E-cadherin expression [55,78].

Finally, we analyzed the possibility of using E-cadherin tumor tissue staining intensity as a diagnostic biomarker to help distinguish ovarian carcinoma from benign ovarian tumors. The results published thus far are not promising, as in most cases, a correlation between the immunoreactivity of this adhesion molecule and tumor malignancy was not found. Hence, there is a high probability that E-cadherin might not be useful in the diagnosis for discriminating malignant and nonmalignant ovarian tumors.

At this point, it is crucial to emphasize some great limitations of E-cadherin expression testing in patients’ tumor tissue, which strongly affect the capability of this analysis to give clear answers. Firstly, there is a lack of a unified method/system for the quantification of protein immunoreactivity. The freedom to determine the percentage of positively stained cells as a cut off value was observed among all analyzed papers. It differed from 1% [55] to 50% [73] of immunoreactive cells. However, ≥10% or ≥25% of stained cells were accepted as positive expression. Secondly, the inclusion or omission of the intensity of staining to create the final semiquantitative scoring system can greatly affect the final interpretation of expression. Thirdly, the source of primary antibodies and their concentration varied substantially between studies, which could also change the staining intensity and alter the final results. However, after in depth analysis of published results, we cannot indicate that there is one single limiting factor that dominantly affects the p value. In our opinion, all general differences in quantification methods of E-cadherin immunoreactivity can collectively influence the results, and it is not possible to point at one leading agent.

## 4. Conclusions

Collectively, we can assume that reduced E-cadherin expression in ovarian cancer tissue is associated with advancement of the disease, especially in the high-grade serous ovarian carcinomas and poor OS of patients. On the other hand, most studies suggest that this protein is not useful for differentiating ovarian carcinoma from benign ovarian tumors.

## Figures and Tables

**Figure 1 ijms-23-14383-f001:**
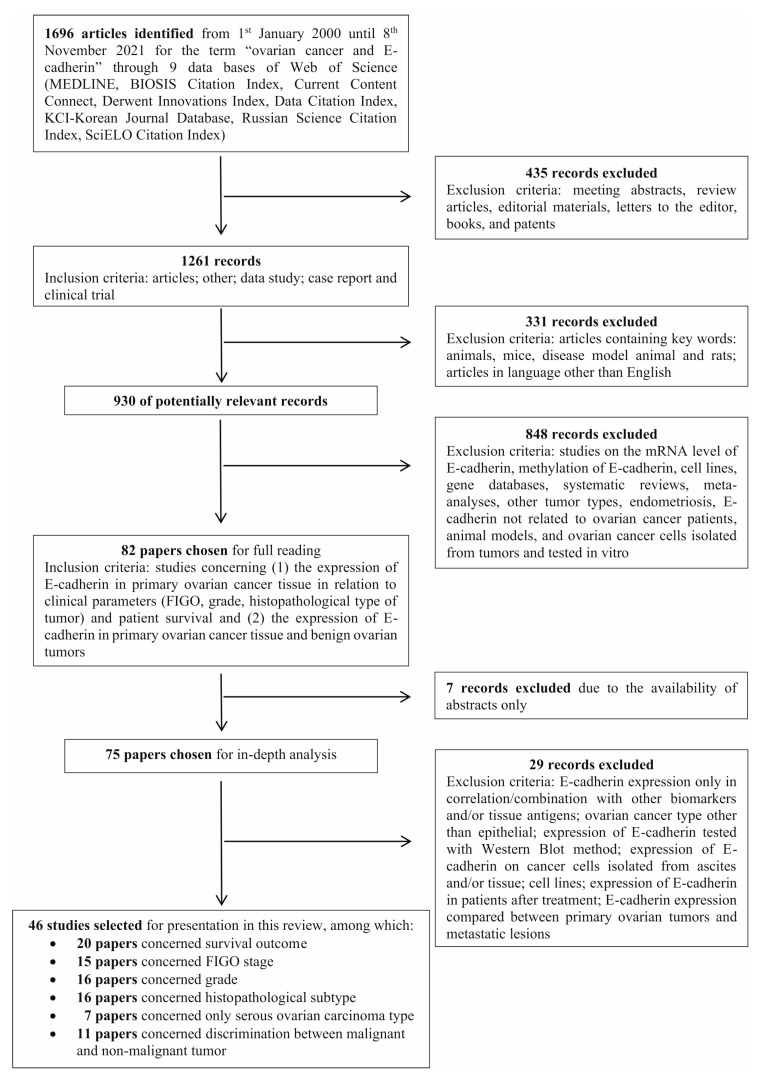
Flow diagram of study selection.

**Table 1 ijms-23-14383-t001:** The association of E-cadherin expression with the OS and PFS of EOC patients.

E-Cadherin Expression	Number of Patients	Follow-Up (Months)	Median OS (Months)	Number of Deaths	*p* Value	HR	95% CI	Localiza-tion	Cutoff [%]	Ref
Preserved Reduced	74 24	140	61 22		<0.001	2.7	1.3–5.9	M	10	[33]
Preserved Reduced	35 19	168	55 23		0.001	3.084	1.541–6.175	M	10	[34]
Preserved Reduced	24 72	175	99 41		0.043	3.084	1.541–6.175	NL	Nd	[35]
Preserved Reduced	72 126	180			0.213	1.245	0.882–1.756	NL	Nd	[36]
Preserved Reduced	177	36	Mean months 27 19		0.008			M	Nd	[37]
Preserved Reduced	14 16	124			<0.001	9.6	2.1–43.6	M + C	1	[38]
Weak * Moderate ** Strong ***	53 41 6	160			0.635 ^(^* ^vs.^ **^)^ 0.103 ^(^* ^vs.^ ***^)^ 0.070 ^(^** ^vs.^ ***^)^			NL	Nd	[39]
Preserved Reduced	36 12	60	48.8 17.9		0.008	2.82	1.3–6.3	NL	<10	[40]
Preserved Reduced	97 7	60		(29%) (66%)	0.006	4.83 (*p* = 0.014)	1.3795–16.9259	M	10	[41]
Preserved Reduced	37 12	200			0.02			M + C	5	[42]
Preserved Reduced	60 63	175		12 (19%) 38 (63%)	0.000			M	10	[43]
Preserved Reduced	46 35	180	52 21		<0.001	1.9–5.8		M	10	[44]
Preserved Reduced	120 16	67			0.6547	1.147	0.629–2.091	NL	6	[45]
Preserved Reduced		148			0.691	1.151 1.00	0.575–2.307	M	10	[46]
Preserved Reduced	34 10		Mean months 79 98		0.491			M	10	[47]
Preserved Reduced	39 7		50 36		0.472			M	10	[48]
			**Mean RFS** **(patients)**							
Preserved Reduced	78 204	120	43 106		0.038			M	5	[49]
			**Median PFS** **(Months)**							
Preserved Reduced	74 24	160	24 9		<0.001	4.3	2.5–7	M	10	[33]
Preserved Reduced	24 72	175	38 17		0.007			NL	Nd	[35]
Preserved Reduced	177	36	23 9		0.000			M	Nd	[37]
Preserved Reduced	14 16				0.064			M + C	1	[38]
Weak * Moderate ** Strong ***	53 41 6	160			0.337 ^(^* ^vs.^ **^)^ 0.679 ^(^* ^vs.^ ***^)^ 0.532 ^(^** ^vs.^ *****^)^			NL	Nd	[39]
Preserved Reduced	42 21	175			0.001			M	10	[43]
Preserved Reduced	34 10		Mean months 42 46		0.967			M	10	[47]
Preserved Reduced	39 7		17 18		0.775			M	10	[48]

Nd—no data; *,**,*** — statistical comparison between groups: weak, moderate and strong in reference [39]; NL—not localized; M—membranous; C—cytoplasmic; M + C—membranous + cytoplasmic (total).

**Table 2 ijms-23-14383-t002:** The association of E-cadherin expression with FIGO stage of EOC patients.

Number of All Patients	FIGO	Number of Preserved/Positive Patients (%)	Number of Reduced/Negative Patients (%)		*p* Value	Localization	Cutoff [%]	Ref.
80	I/II III/IV	≥++ 21 (72) * 20 (39)	+ 6 (21) 16 (31)	− 3 (10) 14 (28)	* <0.05	M	1	[25]
73	I II III IV	19 (70) 6 (60) 13 (39) 1 (33)	8 (30) * 4 (40) 20 (61) 2 (67)		* 0.017 (vs. FIGO IV)	M+C	Nd	[26]
76	I II III IV	13 (48) 5 (50) 9 (25) 0 (0)	14 (52) * 5 (50) 27(75) 3 (100)		* 0.027 (vs. FIGO IV)	M	Nd	[26]
74	I II III IV	2 (7) 1 (10) 3 (9) 0 (0)	25 (93) 9 (90) 31 (91) 3(100)		1.000	C	Nd	[26]
95	I II III IV I/II III/IV	39 (91) 3 (100) 62 (87) 12 (71) 42 (89) 74 (84)	nd		ns	M	10	[32]
123	I/II III/IV		41 (65) 22 (37)		0.002	M	10	[43]
282	I II III IV	27 (35) 10 (23) 36 (27) 5 (19)	56 (65) 34 (77) 98 (73) 22 (81)		0.12	M	5	[49]
50	I/II III/IV	8 (67) 12 (32) *	4 (33) 26 (68) *		0.044	NL	10	[51]
60	I/II III/IV	12 (67) 5 (12)	nd		<0.01	NL	25	[52]
75	I II III IV	116.9 ± 86.6 * 106.1 ± 67.8 ** 101.0 ± 74.8 ** 52.7 ± 15.0	nd		* ≤0.05 ** ≤0.01 (vs. FIGO IV)	C	Nd	[53]
300	I/II III/IV	80 (56) 11 (7)	63 (44) 146 (93)		<0.001	NL	5	[54]
77	I/II III/IV	++ 40 (83) * 17 (59)	+ 8 (17) 10 (34)	− 0 2 (7)	* 0.05	NL	1	[55]
68	I II III IV	++ 10 (27) 3 (30) 9 (50) * 0	+ 19 (51) 5 (50) 6 (33) 1 (33)	− 8 (22) 2 (20) 3 (17) 2 (67)	* <0.01	NL	Nd	[56]
64	I II III IV	nd	5 (42) 2 (40) 12 (27) 0		0.63	M	25	[57]
46	I/II III/IV	14 (70) 15 (53)	6 (30) 13 (46)		0.35	NL	10	[58]

Nd—no data; ns—not significant; “−“ no E-cadherin immunoreactivity; “+” low E-cadherin immunoreactivity; “++” high E-cadherin immunoreactivity; *, **—statistical significance; NL—not localized; M—membranous; C—cytoplasmic.

**Table 3 ijms-23-14383-t003:** The association of E-cadherin expression with FIGO of serous ovarian cancer patients.

Number of All Patients	Serous Type (FIGO)	Number of Preserved/Positive Patients (%)			Number of Reduced/Negative Patients (%)	*p* Value	Localization	Cutoff [%]	Ref.
100	I/II III/IV	+ 9 (60) 44 (52)	++ 3 (20) 37 (45)	+++ 3 (20) 3 (3)	Nd	0.02	NL	Nd	[39]
43	I/II III/IV	12 (75) 8 (30)			4 (25) 19 (70)	0.004	M + C	5	[59]
50	I/II III/IV	IRS 9.1 2.3			Nd	0.001	NL	6	[60]
72	I II III	12 (70.5) 10 (43) 12 (37.5)			Nd	<0.05	NL	5	[61]

Nd—no data; ns—not significant; “+“ low E-cadherin immunoreactivity; “++” medium E-cadherin immunoreactivity; “+++” high E-cadherin immunoreactivity; NL—not localized; M—membranous; C—cytoplasmic; M + C—membranous + cytoplasmic (total).

**Table 4 ijms-23-14383-t004:** The association of E-cadherin expression with tumor grade of EOC patients.

Number of All Patients	Grade	Number of Preserved/Positive Patients (%)		Number of Reduced/Negative Patients (%)	*p* Value	Localization	Cutoff [%]	Ref.
80	G1/G2 G3	≥++ 37 (58) 5 (31)	+ 15 (23) 6 (38)	− 12 (19) 5 (31)	<0.05	M	1	[25]
73	low high	14 (56) 25 (52)		11 (44) 23 (48)	0.808	M + C	Nd	[26]
76	low high	8 (30) 19 (39)		19 (70) 30 (61)	0.464	M	Nd	[26]
74	low high	2 (8) 4 (8)		24 (92) 44 (92)	1.000	C	Nd	[26]
95	G1 G2 G3	11 (100) 19 (91) 47 (78) *		nd	* 0.03	M	10	[32]
54	low high	5 (62) 30 (65)		3 (38) 16 (35)	0.801	M	10	[34]
123	G1 G2 G3	Nd		20 (57) 35 (56.5) 8 (32)	0.063	M	10	[43]
282	G1 G2 G3	14 (38) 35 (35) 29 (20)		23 (62) * 65 (65) * 116 (80)	0.005 (G1 vs. G3; G2 vs. G3)	M	5	[49]
27	low high	9 (33.75 ± 32.71) 18 (30.44 ± 27.42)		nd	1.00	M	Nd	[50]
50	G1/G2 G3	15 (47) 5 (17)		17 (53) 13 (83)	0.186	NL	10	[51]
60	G1/G2 G3	13 (42) 4 (14)		nd	0.016	NL	25	[52]
75	G1 G2 G3	% of positive cells 120.8 ± 83.7 120.7 ± 106.5 144.2 ± 83.2		nd	ns	C	Nd	[53]
300	G1/G2 G3	74 (60) 17 (10)		50 (40) 159 (90)	<0.001	NL	5	[54]
77	G1 G2 G3	++ 27 (77) 20 (77) 10 (63)	+ 8 (23) 6 (23) 4 (25)	− 0 0 2 (12.5)	ns	NL	1	[55]
64	G1 G2 G3	Nd		1 (11) 2 (13) 16 (40)	0.08	M	25	[57]
46	G1 G2 G3	6 (74) 13 (62) 9 (56)		2 (26) 8 (38) 7 (44)	0.47	NL	10	[58]

Nd—no data; ns—not significant; “−“ no E-cadherin immunoreactivity; “+” low E-cadherin immunoreactivity; “++” high E-cadherin immunoreactivity; *—statistical significance; NL—not localized; M—membranous; C—cytoplasmic; M + C—membranous + cytoplasmic (total).

**Table 5 ijms-23-14383-t005:** The association of E-cadherin expression with grade of serous ovarian carcinoma.

Number of All Patients	Serous Type (Grade)	Number of Preserved/Positive Patients (%)			Number of Reduced/Negative Patients (%)	*p* Value	Localization	Cutoff [%]	Ref.
100	G1 (23) G2 (38) G3 (35)	+ 10 (43.5) 19 (50) 23 (65)	++ 10 (43.5) 19 (50) 11 (31)	+++ 3 (13) 0 (0) 1 (3)	Nd	0.001	NL	Nd	[39]
43	Low grade High grade	13 (56.5) 7 (35)			10 (43.5) 13 (65)	0.158	M + C	5	[59]
50	Low grade High grade	IRS 7.4 5.7			Nd	0.269	NL	6	[60]
72	Low grade High grade	23 (85) 11 (25)			Nd	<0.05	NL	5	[61]
52	Low grade High grade	20 (83) 6 (21)			Nd	0.003	NL	25	[62]
93	Low grade High grade	21 (48) 12 (25)			Nd	0.029	NL	25	[63]
452	Low grade High grade	17 (71) 256 (63)			Nd	0.62	M	5	[64]

Nd—no data; ns—not significant; “+“ low E-cadherin immunoreactivity; “++” medium E-cadherin immunoreactivity; “+++” high E-cadherin immunoreactivity; NL—not localized; M—membranous; C—cytoplasmic; M + C—membranous + cytoplasmic (total).

**Table 6 ijms-23-14383-t006:** The association of E-cadherin expression with histopathological type of EOC patients.

Number of All Patients	Histopathological Type	Number of Preserved/Positive Patients (%)		Number of Reduced/Negative Patients (%)	*p* Value	Localization	Cutoff [%]	Ref.
73	Serous mucinous endometrioid clear cell	9 (31) 9 (69) 11 (79) 7 (64)		20 (69) * 4 (31) 3 (21) 4 (36)	* 0.023	M + C	Nd	[26]
76	Serous mucinous endometrioid clear cell	5 (16) 6 (46) 7 (50) 6 (54.5)		27 (84) * 7 (54) 7 (50) 5 (45.5)	* 0.028	M	Nd	[26]
74	Serous mucinous endometrioid clear cell	1 (3) 2 (15) 1 (7) 2 (18)		29 (97) 11 (85) 13 (93) 9 (82)	0.311	C	Nd	[26]
123	Serous mucinous endometrioid clear cell	Nd		28 (44) 16 (67) 6 (58) 13 (52)	0.320	M	10	[43]
282	Serous mucinous endometrioid clear cell	22 (22) 20 (67) 17 (23) 11 (37)		80 (78) * 10 (33) 57 (77) ** 19 (63)	* <0.0005 ** <0.0005	M	5	[49]
300	Serous mucinous endometrioid clear cell	52 (31) 19 (30) 16 (30) 4 (27)		116 (69) 44 (70) 38 (70) 11 (73)	0.982	NL	5	[54]
77	Serous mucinous endometrioid clear cell	Weak 8 0 5 5	Strong 17 (63) 7 (100) 12 (71) 21 (81)	Nd	ns	NL	1	[55]
68	Serous mucinous endometrioid clear cell	Low 9 (39) 3 (43) 8 (47) 12 (57)	High 6 (26) 3 (43) ^$^ 7 (41) 4 (19)	8 (34) 1 (14) 2 (12) 5 (24)	^$^ <0.001	NL	Nd	[56]
64	Serous mucinous endometrioid clear cell	Nd		10 (36) 2 (20) 1 (17) 2 (50)	0.71	M	25	[57]
46	Serous mucinous endometrioid	7 (41) 5 (83) 6 (60)		10 (59) 1 (17) 4 (40)	0.36	NL	10	[58]
73	Serous mucinous endometrioid	IRS 5.29 ± 0.52 4.62 ± 1.00 5.03 ± 0.53		Nd	ns	NL	1	[68]
84	Serous mucinous endometrioid clear cell	1 (4) 8 (61.5) ^#^ 1 (8) 5 (16)		27 (96) * 5 (38.5) 11 (92) 26 (84)	* <0.001 ^#^ <0.001	NL	10	[69]
104	Serous mucinous endometrioid clear cell	10 (27) 15 (40) 3 (8) 8 (22)		46 (69) * 7 (10) 5 (8) 8 (12)	* 0.001	M	1	[70]

Nd—no data; ns—not significant; NL—not localized; M—membranous; C—cytoplasmic; M + C—membranous + cytoplasmic (total). * Reduced expression of E-cadherin significant in serous HP; ** reduced expression of E-cadherin significant in endometrioid HP; ^#^ positive expression of E-cadherin significant in mucinous HP; ^$^ high positive expression of E-cadherin significant in mucinous HP.

**Table 7 ijms-23-14383-t007:** The comparison of E-cadherin expression in EOC versus BOT patients.

Number of Patients	E-Cad Positive Expression n(%)	*p* Value	IRS	*p* Value	Localization	Cutoff [%]	Ref
EOC = 39 BOT = 9			8 12 *	0.024	NL	10	[30]
EOC = 95 BOT = 23	79 (83) 23 (100) *	0.05			M	10	[32]
EOC = 136 BOT = 45	120 (88) ^#^ 7 (16)	<0.0001			NL	6	[45]
EOC = 75 BOT = 23			124.1 ± 92.9 123.8 ± 106.8	ns	C	Nd	[53]
EOC = 68 BOT = 14	52 (76) 10 (71)	ns			NL	Nd	[56]
EOC = 46 BOT = 13	28 (61) 6 (78)	=0.22			NL	10	[58]
EOC = 63 BOT = 7			4.98 ± 0.68 ^#^ 2.71 ± 1.14	≤0.05	NL	1	[68]
EOC = 30 BOT = 30	25 (83) 30 (100) *	<0.05			M	1	[71]
EOC = 31 BOT = 12			1.61 ± 1.17 7.58 ± 2.97 *	<0.01	M	25	[72]
EOC = 78 BOT = 17	25 (32) 5 (29)	=0.062			M	50	[73]

Ns—not significant; nd—no data; NL—not localized; M—membranous; C—cytoplasmic; M + C—membranous + cytoplasmic (total). * significantly higher E-cadherin expression in BOT versus EOC patients; ^#^ significantly higher E-cadherin expression in EOC versus BOT.

## Data Availability

Not applicable.

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
