# Peer review of "E-Cadherin Expression in Relation to Clinicopathological Parameters and Survival of Patients with Epithelial Ovarian Cancer"

_ijms, 2022, doi:10.3390/ijms232214383_

Round 1

Reviewer 1 Report

General

Summary

Authors tried to provide evidences that reduced E-cadherin immune-reactive expression in EOC tissue is associated with less favorable disease outcome, advanced disease, high-grade serous ovarian carcinomas and poor prognosis by literature review. Major concerns are listed below. Authors have tried to present the possible limitations of this review article. The results must be improved to be in line with authors’ conclusions: “we assume that reduced E-cadherin expression in primary ovarian cancer tissue may indicate on less favorable disease outcome and high advancement of the disease. ’’Moreover, E-cadherin expression cannot be considered as a potential biomarker in ovarian cancer….” Hence, a lack of valid or practical information provided in this study is still not convincing.

Title

The methodology has not yet fulfilled the definition and criterion of a systemic review. Hence, the title should be changed to fit the methodology.

Abstract

1. Intro is written well.

1.2. Systemic review….

This paragraph should be moved to method section.

2. Methods

2.1. Literature search and 2.2 Selection criteria and identification of relevant studies.

 The methodology does not yet fulfill the definition and criterion of a systemic review.

3. Results

Authors have addressed the determination of E-cadherin expression was not unequivocal, due to semi-quantitative scoring system based on the intensity and percentage of positively stained cells located on the membrane, cytosol or nucleus by IHC, diverse cut-off value, different used antibodies and selection bias etc. Also, the expression results were presented in less standardized ways. Please improve the presentation of data in Tables.

Tables

Generally, the lack of median OS or PFS between the groups (esp. the very rare cases in negative group) across the selected studies did not make the interpretation convincing. The association between low E-cadherin expression and advanced FIGO stage (limited to stage IV)/high tumor grade/serous histology may be convincing across most of the selected studies.  

Discussion

Page 16, lines 440-441

…, there is a high probability that E-cadherin might not be a useful prognostic for discrimination malignant and nonmalignant ovarian tumors.

Authors please clarify the statement: E-cadherin is a diagnostic or prognostic tool in this setting? Any changes in this setting should be revised throughout this article.

Author Response

Response to the  Reviewer 1

We would like to express our deep thanks to the Reviewer 1 for the revision of our paper and for bringing our attention to some very important matters. We will make our best to clarify all of the Reviewer’s queries and improve our manuscript to make it more clear for the reader.

Summary

Authors tried to provide evidences that reduced E-cadherin immune-reactive expression in EOC tissue is associated with less favorable disease outcome, advanced disease, high-grade serous ovarian carcinomas and poor prognosis by literature review. Major concerns are listed below. Authors have tried to present the possible limitations of this review article. The results must be improved to be in line with authors’ conclusions: “we assume that reduced E-cadherin expression in primary ovarian cancer tissue may indicate on less favorable disease outcome and high advancement of the disease. ’’Moreover, E-cadherin expression cannot be considered as a potential biomarker in ovarian cancer….” Hence, a lack of valid or practical information provided in this study is still not convincing.

We agree with the Reviewer that drawing one, strong and practical conclusion regarding usefulness of E-cadherin as diagnostic factor is extremely difficult on the basis of the presented studies. As we emphasized many times in our manuscript, the methodology and results included in analyzed papers differ much between each study, which makes the comprehensive comparison very hard. As it was mentioned, this fact stands for the limitation of our analysis. Nevertheless, we think that presentation of papers which were published during 20 years can give readers the view on the pross and cons of E-cadherin tissue immunoreactivity in ovarian cancer patients. In our opinion the following assumptions can be made: (i) low E-cadherin expression is associated with high advancement of ovarian cancer especially in the case of serous ovarian cancer type; (ii) low E-cadherin can indicate on poor OS of patient; (iii) determination of E-cadherin expression cannot help to distinguish malignant and non-malignant ovarian tumor. According to the Reviewer’s suggestion, we have revised and changed the conclusions in Abstract and Conclusion sections to be more clear and to be more in line with the presented results. Moreover, we have made some changes in the Results section in accordance to the added cutoff values in all Tables.

Title

The methodology has not yet fulfilled the definition and criterion of a systemic review. Hence, the title should be changed to fit the methodology.

Thank You for pointing out this important matter. We totally agree with the Reviewer that the classical systematic review is more restricted in methodology than what we used. Thus, our article is rather the systematic style literature review, since we applied the PRISMA. According to the Reviewer’s suggestion we changed the title to omit ambiguities. 

Abstract

  1. Intro is written well.

1.2. Systemic review….

This paragraph should be moved to method section.

  1. Methods

2.1. Literature search and 2.2 Selection criteria and identification of relevant studies.

 The methodology does not yet fulfill the definition and criterion of a systemic review.

We have changed the headings of mentioned chapters in the revised version of the manuscript to make it more clear that our article is not classical systematic review.

3. Results

Authors have addressed the determination of E-cadherin expression was not unequivocal, due to semi-quantitative scoring system based on the intensity and percentage of positively stained cells located on the membrane, cytosol or nucleus by IHC, diverse cut-off value, different used antibodies and selection bias etc. Also, the expression results were presented in less standardized ways. Please improve the presentation of data in Tables.

Thank You for this comment. According to the suggestion of the Reviewer we added the cutoff values into all tables. The information regarding membrane or cytosol immunoreactivity were already included in our paper. However, the association of positive cells in cytosol or membrane with p value is not noticeable. We value the Reviewer’s remark but we have decided not to attach the information regarding the types of used antibodies or detailed description of IHC method for each study. We simply believe that inclusion of this data would make the tables unreadable and harder to follow. All tables have references for each study and it is easily possible for the reader to track and check the details of particular method.

Tables

Generally, the lack of median OS or PFS between the groups (esp. the very rare cases in negative group) across the selected studies did not make the interpretation convincing. The association between low E-cadherin expression and advanced FIGO stage (limited to stage IV)/high tumor grade/serous histology may be convincing across most of the selected studies.  

Thank You for pointing this out. In Table 1 the median OS or PFS data were included if they were available. Unfortunately, not all analyzed articles contained such information. Some authors provided the mean values or simply percentages of death (as indicated in Table 1). However, since all analyzed papers were  accepted for publication in peer-review journals we decided to include them in Table 1 regardless of the content of median values. Moreover, in the group of articles without median OS or PSF values, 4 out of 7 indicated a significant association between E-cadherin level and patients’ survival. We absolutely agree with the Reviewer that interpretation of data provided by authors of selected publications is not easy and does not allow for straightforward conclusion. Thus, we tried to show how complex the research on E-cadherin are and we tried to point out the most convincing results. On the other hand, we are well aware that on the basis of published papers it is not possible to draw one strong conclusion. Similarly to the Reviewer, we also think that there is an association between the reduced E-cadherin expression and high advancement of disease especially in the group of serous carcinoma. We have made some correction in the Result section to underline this conclusion and make it more clear.

Discussion

Page 16, lines 440-441

…, there is a high probability that E-cadherin might not be a useful prognostic for discrimination malignant and nonmalignant ovarian tumors.

Authors please clarify the statement: E-cadherin is a diagnostic or prognostic tool in this setting? Any changes in this setting should be revised throughout this article.

We would like to thank the Reviewer for bringing our attention to this important matter. Indeed we have lacked the precision in using some of the terms. Our intention is was to analyze the expression of E-cadherin as a diagnostic biomarker. We have revised and corrected the whole manuscript and clarified the indication of E-cadherin as diagnostic tool.

Reviewer 2 Report

The authors have conducted a systematic review to examine the role E-cadherin protein expression in ovarian cancer.  Overall the findings have been presented in a confusing manner. 46 studies were selected for the review but the number of actual studies included in tables wasn’t clear. Some additional studies were excluded but this information was not included in Figure 1.   The HGSOC should be considered separately.  It wasn’t clear if only high serous cancers were included in Table 3.  The key findings need to be presented more clearly.  

Abstract

‘In conclusion, we assume that reduced E-cadherin expression in primary ovarian cancer tissue may indicate on less favorable disease outcome and high advancement of the disease’.”Remove ‘we assume that’ from the sentence.

 Lines 76-80. ‘In addition to processes that regulate E-cadherin gene expression, there is also a great variety of mechanisms focused on posttranslational modification of the E-cadherin protein. One of them includes endocytosis and proteolytic processing as an alternative way of inhibiting the regular function of E-cadherin.’ Under normal conditions, this adhesion molecule undergoes clathrin-dependent endocytosis and is then recycled to a new site in the plasma membrane to form new cell–cell contacts. add refs to these sentences  

 Lines 84-87 ‘Moreover, various metalloproteinases (MMP3, MMP7, MMP9, MMP14, ADAM10) cause shedding of the E-cadherin extracellular domain near the plasma membrane by proteolytic degradation of adherent junctions’ add ref to this sentence

 Lines 117-118 ‘Moreover, our review is a strong voice in the worldwide discussion about the opportunity to consider E-cadherin expression loss as an independent biomarker in ovarian cancer. Sentence needs modification- strong voice is not the correct term to use. Maybe good consensus?

 Figure 1 should also include the number of studies evaluated with survival outcome (table 1), stage (table 2), serous subtype (table 3), grade (table 4), histological subtype (table 5)  

Line 195-197 ‘A total of twenty papers considered the relationship of patient survival with positive and/or negative expression of E-cadherin Only  9 studies are included in table 1.  Is not clear what happened to the other studies ?  

Table 1 should include the cuff off used for the assessment and also details of the assessment method and the antibody used for immunohistochemistry

1st Column should be E-cadherin high or low with the cut point used in the study. The terms Positive (preserved) or Negative (reduced) are confusing

Line 195 ‘A total of twenty papers considered the relationship of patient survival with positive and/or negative expression of E-cadherin’ negative E-cadherin expression should be low E-caherin expression

 Line 222. ‘In summary, 5 out of 7 papers (71%) clearly indicate that preserved expression of E-cadherin is associated with favorable OS of ovarian cancer patients’. Line 249 ‘In summary, most studies (6 out of 10; 60%) support the assumption that reduced/lost expression of E-cadherin results in shorter survival of EOC patients. There are 9 studies included in table 1 not 10 ?

 Table 3. The association of E-cadherin expression with FIGO or tumor grade of serous ovarian cancer patients. Please confirm whether this also included low grade serous cancers. High grade serous cancers should be examined separately

Author Response

Response to the Reviewer 2

We would like to express our grattitude to the Reviewer 2 for his revision of this paper and for pointing out some very interesting and important issues. We will make our best to address all of them, thus improving our manuscript and make it more clear for the reader.

The authors have conducted a systematic review to examine the role E-cadherin protein expression in ovarian cancer.  Overall the findings have been presented in a confusing manner. 46 studies were selected for the review but the number of actual studies included in tables wasn’t clear. Some additional studies were excluded but this information was not included in Figure 1.   The HGSOC should be considered separately.  It wasn’t clear if only high serous cancers were included in Table 3. The key findings need to be presented more clearly.   

Thank You for this opinion. We are aware that some parts of our manuscript may be confusing or unclear, especially regarding to the number of analyzed studies in particular table. We have revised our paper and clarified the description of studies included in the tables. Moreover, we have also clarified Table 3 and improved presentation of key findings.

‘In conclusion, we assume that reduced E-cadherin expression in primary ovarian cancer tissue may indicate on less favorable disease outcome and high advancement of the disease’.”Remove ‘we assume that’ from the sentence.

We have changed the sentence, according to the Reviewer suggestion.

 Lines 76-80. ‘In addition to processes that regulate E-cadherin gene expression, there is also a great variety of mechanisms focused on posttranslational modification of the E-cadherin protein. One of them includes endocytosis and proteolytic processing as an alternative way of inhibiting the regular function of E-cadherin.’ Under normal conditions, this adhesion molecule undergoes clathrin-dependent endocytosis and is then recycled to a new site in the plasma membrane to form new cell–cell contacts. add refs to these sentences.

Lines 84-87 ‘Moreover, various metalloproteinases (MMP3, MMP7, MMP9, MMP14, ADAM10) cause shedding of the E-cadherin extracellular domain near the plasma membrane by proteolytic degradation of adherent junctions’ add ref to this sentence.

Thank You for these comments. The references for these fragments were already included in the manuscript. However, since it may be hard to determine which articles refer to them, we have revised our manuscript and corrected these part so that the articles referring to them are cited more properly. The references have been added to this part, according to the Reviewer suggestion. 

 Lines 117-118 ‘Moreover, our review is a strong voice in the worldwide discussion about the opportunity to consider E-cadherin expression loss as an independent biomarker in ovarian cancer. Sentence needs modification- strong voice is not the correct term to use. Maybe good consensus?

We have modified the sentence, according to the Reviewer suggestion.

 Figure 1 should also include the number of studies evaluated with survival outcome (table 1), stage (table 2), serous subtype (table 3), grade (table 4), histological subtype (table 5).

Thank You for pointing this out. Indeed such an information would be very helpful for the reader. Thus we have revised Figure 1 and added the guidance how many studies were included in each particular analysis. This should make our paper easier to follow.

Line 195-197 ‘A total of twenty papers considered the relationship of patient survival with positive and/or negative expression of E-cadherin’  Only  9 studies are included in table 1.  Is not clear what happened to the other studies? 

Thank You for this important comment. Firstly, we analyzed 20 studies concerning E-cadherin expression and patients survival. Three of them was rejected thus remained 17 papers. In this group two papers referred to patients in FIGO III/IV stage only. Three studies referred to HGSOC patient only. Another two articles described serous OC patients but in all FIGO stages.  These 7 out of 17 studies were described in text lines 198-231, however we did not included these studies in Table 1, since they were not referred to EOC patients in all FIGO stages and all HP types. 9 article determined the association of E-cadherin expression and OS of EOC patients representing all stages, grades and histopathological types and these studies were included in Table 1. Within these 9 papers, 3 referred to PFS in EOC patients representing all stages, grades and histopathological types. One las paper presented calculated RFS and this data we described in text only. Therefore, in Table 1 we eventually included 9 out of 20 studies.

Table 1 should include the cuff off used for the assessment and also details of the assessment method and the antibody used for immunohistochemistry.

We thank the Reviewer for this valuable comment. According to Reviewer suggestion, we have added the cut off values to the Table 1 and all others. However, decided not to attach the information regarding the types of used antibodies or detailed description of IHC method for each study. We believe that inclusion of this data would make the tables unreadable and harder to follow. All tables have references for each study and it is easily possible for the reader to track and check the details of particular method.

1st Column should be E-cadherin high or low with the cut point used in the study. The terms Positive (preserved) or Negative (reduced) are confusing.

Line 195 ‘A total of twenty papers considered the relationship of patient survival with positive and/or negative expression of E-cadherin’ negative E-cadherin expression should be low E-cadherin expression.

Thank Your for bringing these two points up. We understand that using double terms: “Positive (preserved)” or “Negative (reduced)” may be confusing and may be not giving the perfect representation of actual E-cadherin level. However, we would like to inform that such terms were used by the authors of presented studies and it just shows how variable the methodology of all included papers was. We would like to simplify this nomenclature and make it more clear, however we cannot really use the terms “high expression of E-cadherin” or “low expression of E-cadherin”, since cut of values differs substantially between studies, with some of them being very low (ex. 1%), while others being very high (ex. 50%). Therefore it is not really appropriate to qualify a positive result as “high expression” simultaneously for extremely distant cut off values (like 1% and 50%). Our proposition is to keep the terms of “preserved” and “reduced” E-cadherin, as a consensus and simplification of nomenclature.

 Line 222. ‘In summary, 5 out of 7 papers (71%) clearly indicate that preserved expression of E-cadherin is associated with favorable OS of ovarian cancer patients’. Line 249 ‘In summary, most studies (6 out of 10; 60%) support the assumption that reduced/lost expression of E-cadherin results in shorter survival of EOC patients. There are 9 studies included in table 1 not 10 ?

Thank You for this point. The detailed clarification regarding the number of papers used in Table 1 is described above.

 Table 3. The association of E-cadherin expression with FIGO or tumor grade of serous ovarian cancer patients. Please confirm whether this also included low grade serous cancers. High grade serous cancers should be examined separately

Thank you for your comment. In the reviewed version we have divided Table 3 into two separate tables (3 and 5) and now they are presenting the association of E-cadherin expression with FIGO of serous ovarian cancer (Table 3) and the association of E-cadherin expression with tumor grade of serous ovarian cancer (Table 5). We confirm, that in the revised table 3 serous ovarian carcinoma include both, low (LGOSC) and high grades (HGOSC). However, Authors of these papers did not analysed FIGO stage referring to LGOSC and HGOSC and neither they did provide the exact number of LGOSC and HGOSC.

New Table 5 contains data regarding E-cadherin expression in serous ovarian carcinoma in association with low or high tumor grade (as shown in the table).   

Round 2

Reviewer 1 Report

Thanks for authors’ efforts to improve this article. Nevertheless, I look forward to further improvement and further presentation to be clarified:

Page 5

The “Fig. 1” text below the algorithm (prior to the legend of Figure 1) should be removed.

Table 1

The p values should be clarified: “0.000” [Ref 43;37] and “0.07””0.337” [Ref 39].

Author Response

Response to Reviewer 1

We would like to express our gratitude to the Reviewer 1 for the second round of revision of our paper and for pointing out some unclear matters. We will make our best to clarify all of the Reviewer’s queries.

Page 5

The “Fig. 1” text below the algorithm (prior to the legend of Figure 1) should be removed.

Thank You for brining our attention to this matter.  According to the Reviewer’s suggestion we have removed “ Fig 1” text on the Page 5. The Microsoft tracking system removed the Figure 1 from the first round of revision and now it is replaced with the final, proper figure (thus the previous Figure is not visible).

Table 1

The p values should be clarified: “0.000” [Ref 43;37] and “≥0.07””≥0.337” [Ref 39].

We would like to thank the Reviewer for pointing this out. The p values presented in Table 1 were taken directly from the articles referring to them. The p values of 0.000 were taken from paper of Kim K, et all 2014 [43] and Quattrocchi L, et al 2011 [37]. The tables presented in these manuscripts consisted of  p values with 3 digits after decimal point even though in some cases these values were so low that these digits were only 0. As for the p value taken from paper of  Dian D, et all 2011 [39] indeed we have taken only the lowest p value of each analysis without detailed distinction of particular comparison of each group. In the revised version of Table 1 we have included all the p values for statistical comparison of every group presented in the study (eg. Weak vs Moderate, Weak vs Strong, Moderate vs Strong).

Reviewer 2 Report

The manuscript has been  improved  and now suitable for publication 

Author Response

Response to Reviewer 2

We would like to express our gratitude to the Reviewer 2 for his revision of this paper. We are delighted to see that our responses and explanation of round 1 comments were satisfying for the Reviewer 2.